# Improving End-to-End Speech Processing by Efficient Text Data Utilization with Latent Synthesis

**Jianqiao Lu**[1*†]**, Wenyong Huang**[2*]**,**
**Nianzu Zheng**[2]**, Xingshan Zeng**[2]**, Yu Ting Yeung**[2] **& Xiao Chen**[2]
[1]The University of Hong Kong    [2]Huawei Noah's Ark Lab
jqlu@cs.hku.hk, wenyong.huang@huawei.com

## Abstract

Training a high performance end-to-end speech (E2E) processing model requires an enormous amount of labeled speech data, especially in the era of data-centric artificial intelligence. However, labeled speech data are usually scarcer and more expensive for collection, compared to textual data. We propose Latent Synthesis (LaSyn), an efficient textual data utilization framework for E2E speech processing models. We train a latent synthesizer to convert textual data into an intermediate latent representation of a pre-trained speech model. These pseudo acoustic representations of textual data augment acoustic data for model training. We evaluate LaSyn on low-resource automatic speech recognition (ASR) and spoken language understanding (SLU) tasks. For ASR, LaSyn improves an E2E baseline trained on LibriSpeech train-clean-100, with relative word error rate reductions over 22.3% on different test sets. For SLU, LaSyn improves our E2E baseline by absolute 4.1% for intent classification accuracy and 3.8% for slot filling SLU-F1 on SLURP, and absolute 4.49% and 2.25% for exact match (EM) and EM-Tree accuracies on STOP respectively. With fewer parameters, the results of LaSyn are competitive to published state-of-the-art works. The results demonstrate the quality of the augmented training data.

## 1 Introduction

In the data-centric artificial intelligence era, large quantity and high quality training data are essential for good performance of natural language processing (NLP) models including speech processing models. A conventional speech processing system is usually cascaded with an automatic speech recognition (ASR) module and an NLP module. For example, in spoken language understanding (SLU) which predicts semantic information from speech input, the system first transcribes input speech into

text with ASR, then pipes the text output to the natural language understanding (NLU) model for text analysis. An end-to-end (E2E) speech processing system leverages a single model which takes the input speech and performs spoken language processing tasks simultaneously. E2E models draw increasing attention due to less computational complexity and error propagation mitigation (Shen et al., 2021; Tian and Gorinski, 2020; Sharma et al., 2021; Lugosch et al., 2020; Wang et al., 2020; Chen et al., 2021b). However, a challenge of E2E model training is the collection of enormous annotated spoken data, which are significantly more expensive to collect compared with the text-only counterpart. In contrast, for a cascaded system, the ASR module and NLP module are trained separately with paired speech-transcription data and annotated textual data respectively. Separated types of data are usually more readily available and thus lower data collection costs. As the amount of high quality training data is critical for an E2E model, a strategy to alleviate the inadequate spoken data problem with more abundant textual data.

Two approaches have been proposed for utilizing textual data for E2E speech models in the literature. The first is modality conversion which utilizes a text-to-speech (TTS) system to convert text into speech (Laptev et al., 2020). The disadvantage is the requirement for a high-quality expressive TTS system. Another approach is unified representation learning for matching latent representations of speech and text with alignment losses (Bapna et al., 2021; Chen et al., 2022a). Given the significant difference between speech and text, aligning the hidden latent space of the two modalities is challenging.

We propose Latent Synthesis (LaSyn), a method to utilize text-only data for E2E speech processing models. LaSyn can be seen as an integration of the above two ideas. We train a latent synthesis model which synthesizes textual data into an intermediate

---

*Leading co-authors with equal contribution.
†Work done during an internship at Huawei.

latent representation of a pre-trained speech model. Compared to modality conversion, speech latent representation contains fewer details and redundancy than the original speech signal, thus is easier to synthesize. Compared to unified representation learning, instead of aligning two modalities of huge difference, LaSyn learns to map the text into the latent representation of speech directly.

We evaluate LaSyn on low-resource ASR and SLU tasks. Low-resource ASR has gained big progress with the advancement of self-supervised speech pre-training (Baevski et al., 2020; Hsu et al., 2021; Huang et al., 2022). Further performance improvement still relies on external language models (Baevski et al., 2020). We show that LaSyn allows an E2E ASR model to utilize text-only data effectively without external language models, and outperforms ASR models with external language models. We further evaluate LaSyn on two publicly available datasets for SLU tasks, namely SLURP (Bastianelli et al., 2020) and Spoken Task Oriented Semantic Parsing (STOP) (Tomasello et al., 2022). LaSyn achieves comparable performance to the state-of-the-art (SOTA) SLU models but with significantly fewer model parameters. We summarize our contributions as follows:

- We propose LaSyn, an efficient textual data utilization framework for E2E speech processing models. The framework enables cross-modal knowledge transfer from text to E2E speech processing models through latent synthesis.

- We design 2 implementations for latent synthesizer which is the core of LaSyn framework: a fixed-projection latent synthesizer, and a diffusion latent synthesizer which applies recent progress of generative model, diffusion probabilistic model (Ho et al., 2020; Song et al., 2020).

- By improving an E2E ASR model through textual data utilization with LaSyn, we achieve competitive results on a low-resource ASR setup than published supervised ASR models which utilize textual data through an external language model.

- With LaSyn, we demonstrate E2E SLU models can be improved with a diverse set of textual NLP tasks, including NLU, information extraction (IE), named entity recognition (NER), and masked language modeling (MLM). We achieve competitive results to published SOTA works on two publicly available SLU datasets, with significantly fewer model parameters.

This paper is organized as follows. In the next section, we discuss related works of LaSyn. In Section 3, we discuss the model structure and training of LaSyn. We present experimental setup and results in Section 4, and ablation studies on SLU tasks in Section 5. Finally, we conclude our work in Section 6.

## 2 Related Works

In this section, we discuss the prior works of modality conversion and unified representation learning related to LaSyn.

**Modality conversion:** Laptev et al. (2020) shows that TTS data augmentation improves ASR performance in a low-resource setting. Sun et al. (2020) further shows that the diversity and quality of the TTS system are important for ASR data augmentation. Chen et al. (2022b) demonstrates similar representations derived from synthesized speech help downstream ASR tasks. Lugosch et al. (2020) confirms the effectiveness of speech synthesis for E2E SLU models, either as a sole source of training data or as a form of data augmentation. Thomas et al. (2021) utilizes artificially synthesized speech to adapt a SLU model based on a recurrent neural network transducer. Huang et al. (2020b) demonstrates the effectiveness of a multi-speaker TTS system under a low-resource SLU setting. Kharitonov et al. (2023) decouples the text-to-semantic and semantic-to-acoustic tasks to realize a multi-speaker text-to-speech system. LaSyn generates pseudo acoustic representations from text without requiring a vocoder for speech waveform generation.

**Unified representation learning:** Ao et al. (2021) extends the idea of T5 (Raffel et al., 2020) and proposes Speech-T5 with a cross-modal vector quantization in a shared discrete latent space. Kim et al. (2021) learns multi-modal alignment with two cross-modal pre-training tasks of masked language modeling and conditioned language modeling. Qian et al. (2021) unifies a pre-trained ASR encoder for speech and a pre-trained language model encoder for text into a transformer decoder. Sato et al. (2022) introduces an adaptation branch to embed acoustic and linguistic information in the same latent space. Thomas et al. (2022) trains an RNN-T model both on speech and text inputs. Zhang et al. (2022a) introduces two alternative discrete phoneme-unit and hidden-unit tokenizers to

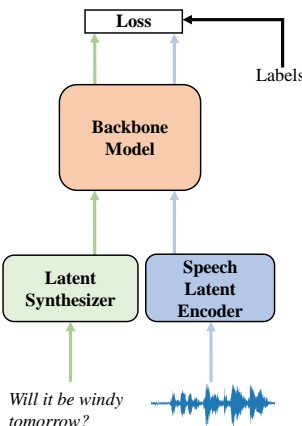

Figure 1: The architecture of LaSyn framework.

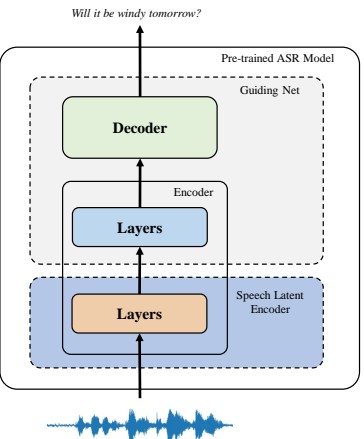

Figure 2: Speech Latent Encoder and Guiding Net from a pre-trained ASR model.

bridge speech and text modalities. MAESTRO (Chen et al., 2022a) learns unified representations of text and speech through sequence matching and duration prediction. Chung et al. (2018) attempts to align the individually learned text and speech embedding via adversarial training and a refinement procedure. SpeechUT (Zhang et al., 2022b) leverages hidden units as the bridge between the speech encoder and the text decoder. SpeechGPT(Zhang et al., 2023) applies modality-adaptation pertaining and cross-modal instruction fine-tuning to perceive and generate multi-model content. LaSyn connects text and speech information by mapping text representation directly into the pseudo acoustic latent space of a pre-trained speech model.

## 3  Method

### 3.1  Architecture

The LaSyn framework is illustrated in Fig. 1. The framework has 3 components: a speech latent encoder which maps speech data to corresponding speech latent representation, a latent synthesizer that projects text into the speech latent space, and a backbone model which is trained with either speech latent representations or pseudo acoustic latent representations from text.

### 3.2  Training procedure

#### 3.2.1  Speech Latent Encoder

Speech latent encoder is obtained from a pre-trained speech processing model, which is a supervised ASR model as illustrated in Fig. 2 in this work. The parameters of speech latent encoder are frozen in the latter training stages to fix the speech latent space.

#### 3.2.2  Latent Synthesizer

We then train a latent synthesizer to project textual data into the same speech latent space of the speech latent encoder. Latent synthesizer allows utilizing training samples from textual data, which is the core of the LaSyn framework. We explore two implementations of the latent synthesizer.

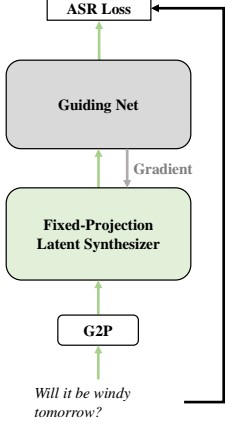

Figure 3: Training process of Fixed-Projection Latent Synthesizer. We freeze parameters of Guiding Net.

**Fixed-projection Latent Synthesizer:** We train a fixed-projection latent synthesizer with the help of a guiding net. The guiding net is also obtained from the pre-trained ASR model as illustrated in Fig. 2. Note that the guiding net is frozen in this stage. The training procedure is illustrated in Fig. 3. We optimize a fixed-projection latent synthesizer to generate latent representations which are recognizable as input of the guiding net. As the name suggests, the fixed-projection latent synthesizer learns a fixed one-to-one projection between

text data and speech latent representation. The training objective is defined as follows,

$$\underset{\phi}{\arg\min}\ \mathcal{L}_{ASR}\Big(G_\theta\big(P_\phi(\text{G2P}(t))\big), t\Big) \qquad (1)$$

where $G_\theta$ and $P_\phi$ represent the guiding network and the fixed-projection latent synthesizer respectively, $\phi$ represents the parameters of the latent synthesizer, $t$ is the text input, and G2P is a grapheme-to-phoneme module. $\mathcal{L}_{ASR}$ is the same loss function of the pre-trained ASR model, such as transducer loss (Graves, 2012) or cross-entropy loss for attention-based encoder-decoder (AED) (Vaswani et al., 2017).

**Diffusion Latent Synthesizer:** We also experiment with diffusion probabilistic models (DPM) (Ho et al., 2020) as the latent synthesizer. DPMs have achieved great success in TTS (Popov et al., 2021; Chen et al., 2021a) and text-conditioned image synthesis (Nichol et al., 2021; Saharia et al., 2022) recently. We use the formulation of DPM proposed in Karras et al. (2022). Diffusion latent synthesizer generates latent representations by sampling an initial latent representation from a noise distribution and iteratively denoising the sample using a denoising model $D(h_{noisy}; \boldsymbol{e}, \sigma)$ where $h_{noisy}$ represents the noisy latent at the current step, $\boldsymbol{e}$ denotes the conditional text. The denoising model is composed of an UNet (Ronneberger et al., 2015) and a text encoder as shown in Fig. 4. To reduce the complexity of the diffusion model, we train an autoencoder to compress the latent representation and use the lower-dimensional latent representation as the target of the diffusion latent synthesizer, similar to Rombach et al. (2022). For succinctness, we do not depict the training of autoencoder in Fig. 4. The training objective is to minimize,

$$\mathbb{E}_{p(h,\boldsymbol{e}),p(\boldsymbol{\epsilon}),p(\sigma)}\left[\lambda(\sigma)\big\|D(h+\sigma\boldsymbol{\epsilon};\boldsymbol{e},\sigma)-h\big\|_2^2\right] \tag{2}$$

where $h$ is clean latent representation, $p(h, \boldsymbol{e})$ represents the training data distribution of latent-text pairs. The latent-text pairs are derived from a paired speech-text dataset and a speech latent encoder which converts the speeches into latent representations. $p(\sigma)$ is the distribution of noise levels that defines the corruption schedule (Karras et al., 2022). $p(\boldsymbol{\epsilon}) \in \mathcal{N}(\boldsymbol{0}, \boldsymbol{1})$ is the standard normal

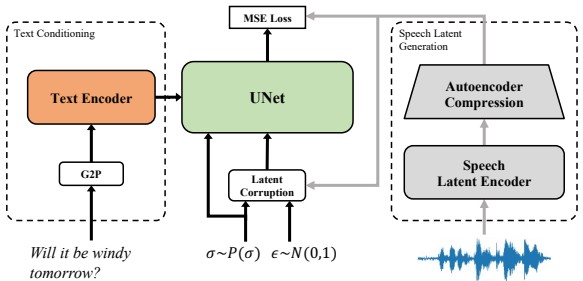

Figure 4: Diffusion Latent Synthesizer training. The gray color indicates that the Speech Latent Encoder and Autoencoder are frozen during training.

distribution, $\lambda(\sigma)$ is the weighting factor of noise levels. We employ classifier-free diffusion guidance (Ho and Salimans, 2022) to control latent quality and text alignment when sampling from the diffusion latent synthesizer.

### 3.2.3 Backbone Model and Dual-modality Training

After we train the latent synthesizer, we train the backbone model. We freeze the speech latent encoder and the latent synthesizer during backbone model training. We utilize both text and speech data in training. The backbone model takes input latent features from either speech latent encoder or latent synthesizer. We formulate both text-to-text and speech-to-text tasks as a unified sequence-to-sequence problem and refer to as dual-modality training. The training loss is specific to each task, i.e., transducer loss for ASR, and cross-entropy loss for SLU. The amount of textual data is usually significantly larger than speech data. We first train the backbone model with textual data. Then we train the backbone model with both text and speech data.

## 4 Experiments

### 4.1 Training Data

#### 4.1.1 ASR

We apply a 100-hour subset (train-clean-100) of LibriSpeech (Panayotov et al., 2015) as low-resource labeled speech data. We use the transcription of the whole 960-hour LibriSpeech training split (LS-960) as text-only data.

#### 4.1.2 SLU

We evaluate LaSyn on two challenging SLU datasets, SLURP (Bastianelli et al., 2020) and

| Task | Dataset |
|------|---------|
| NLU | CLINC150 (Larson et al., 2019) |
| | Redwood (Larson and Leach, 2022) |
| | GOOGLE-DSTC8 (Rastogi et al., 2020) |
| | Leyzer (Sowański and Janicki, 2020) |
| | HINT3 (Arora et al., 2020) |
| | Chatbot-Corpus (Braun et al., 2017) |
| | MultiWOZ (Zang et al., 2020) |
| | BANKING77 (Casanueva et al., 2020) |
| | FEWSHOTWOZ (Peng et al., 2020) |
| | ATIS (Tur et al., 2010) |
| | Schema (Rastogi et al., 2019) |
| NER | CrossNER (Liu et al., 2020) |
| | WNUT17 (Derczynski et al., 2017) |
| | CoNLL-2003 (Tjong Kim Sang and De Meulder, 2003) |
| | CoNLL-2004 (Carreras and Màrquez, 2004) |
| IE | OntoNotes (Weischedel et al., 2013) |
| | SCIERC (Luan et al., 2018) |

Table 1: Extra NLP datasets for SLU experiments.

| | |
|---|---|
| Channel multiplier | [1, 1, 1, 1] |
| Dropout | 0.1 |
| Number of channels | 256 |
| Number of residual blocks | 1 |
| Self attention resolutions | [4, 2] |

Table 2: Hyper-parameters of UNet model

STOP (Tomasello et al., 2022). SLURP is substantially larger and linguistically more diverse than previous SLU datasets. STOP is a recently released dataset that is the largest and the most complex SLU dataset. We also leverage a diverse set of NLP text datasets from different tasks, including natural language understanding (NLU), named entity recognition (NER), and information extraction (IE). The extra NLP text datasets are listed in Table 1.

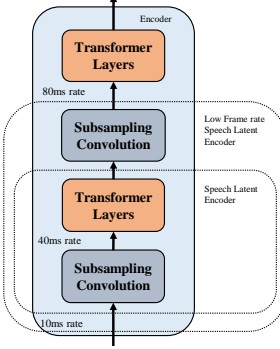

Figure 5: Encoder architecture of the ASR model. The frame rate of input is denoted as '10/40/80 ms'.

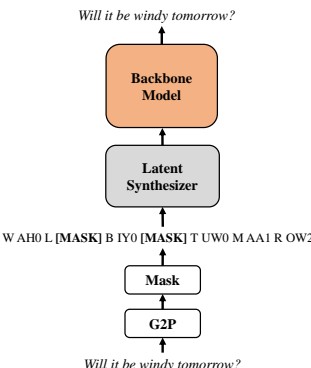

Figure 6: MLM task for utilizing unlabeled text data. [MASK] denotes the masked position.

## 4.2 Model and Training Setups

### 4.2.1 ASR

For ASR pre-training, we use a Transformer Transducer model (Tian et al., 2019; Yeh et al., 2019; Zhang et al., 2020). We apply a 128-dimensional log-mel filterbank with 20 ms window length and 10 ms frame rate as input acoustic feature. We interleave strided-convolutions in the encoder to gradually down-sample the input speech as illustrated in Fig. 5, which reduces computation effectively with negligible performance degradation (Peddinti et al., 2018; Han et al., 2020; Huang et al., 2020a). This model is pre-trained with train-clean-100. SpecAugment (Park et al., 2020) is applied to avoid overfitting. This pre-trained model is also our E2E ASR baseline. We obtain a speech latent encoder from this pre-trained model.

For latent synthesizers, we evaluate both fixed-projection latent synthesizer and diffusion latent synthesizer. The fixed-projection latent synthesizer is composed of 4 1-D convolutional layers of 512 filters with a kernel size of 5. We observe that a simple model structure is sufficient. We train the diffusion latent synthesizer with train-clean-100. The text encoder is composed of two convolution layers followed by the two-layer transformer. The number of channels is 256. The UNet model is adapted for 1-D sequence processing. The hyper-parameters of the UNet model are listed in Table 2. We use a small model such that the latent synthesizer generates the pseudo acoustic latent representations on the fly during dual-modality training.

The backbone model is the same as the guiding net in Fig. 2. To utilize textual data in dual-modality training of the backbone model, we design a task similar to masked language modeling

(MLM) ([Devlin et al., 2018](#)) as illustrated in Fig. 6. We randomly mask 30% of input phonemes converted by g2pE[1] according to CMUDict[2], and train the backbone model to predict the corresponding words.

We note that the parameters of the guiding net are frozen in latent synthesizer training. If we do not provide textual data for backbone model training, we just update the E2E baseline with extra epochs with a frozen speech latent encoder.

### 4.2.2 SLU

We apply an attention-based encoder-decoder model for ASR pre-training. The pre-trained ASR model is trained with LS-960 and SLURP speech data. The structure of the encoder is similar to the one in ASR experiments described in section 4.1.2. We apply a 6-layer, 256-dimensional Transformer as the decoder. We evaluate the two implementations of the latent synthesizer similar to ASR experiments. For fixed-projection latent synthesizer, the configuration is the same as ASR experiments. We apply text transcription of LS-960 for training. For diffusion latent synthesizer, we use LS-960 as paired speech-text training data. The backbone model shares the same model structure as the guiding net in Fig. 2. We also initialize the parameters from the guiding net. We train the backbone model with multiple tasks, including SLU, NLU, NER, and IE. We convert the annotation of all the datasets to a text-sequence format as illustrated in Fig. 7. We formulate all the tasks as a unified sequence-to-sequence problem.

We note that the model structure of the E2E baseline model is the same as the LaSyn model, but the latent synthesizer is disabled. The E2E baseline model does not train with any additional textual data. We fine-tune the E2E baseline model with SLU task after ASR pre-training.

### 4.3 ASR Results

The experimental results of ASR are shown in Table 3. We first compare LaSyn models with our E2E baseline which achieves comparable performance to conformer-based models. The only difference is that the LaSyn models are trained with additional textual data. The LaSyn-Diffusion model, which uses a diffusion latent synthesizer, achieves 40.5% and 22.3% relative WER reductions on test-clean and test-other of Librispeech test sets com-

[1]https://github.com/Kyubyong/g2p
[2]https://github.com/cmusphinx/cmudict

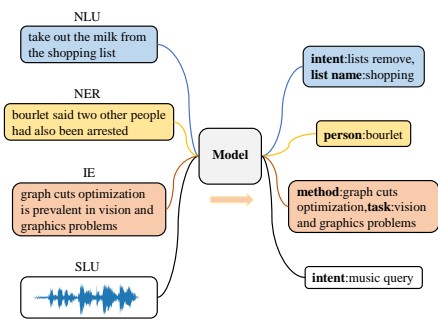

Figure 7: Dual-modality training for SLU with LaSyn. The output labels of different tasks are converted to text sequences as shown in the right blocks. Meta values such as slot type and entry type are in bold.

| Model | LM | test | |
| --- | --- | --- | --- |
| | | clean | other |
| Hybrid DNN/HMM ([Lüscher et al., 2019](#)) | 4-gram | 5.8 | 18.6 |
| LAS ([Park et al., 2020](#)) | LSTM | 5.5 | 16.9 |
| Conformer-CTC ([Watanabe et al., 2022](#)) | - | 7.7 | 20.6 |
| Conformer-CTC/Attention ([Watanabe et al., 2022](#)) | - | 7.3 | 19.3 |
| Conformer-Transducer ([Watanabe et al., 2022](#)) | - | 7.8 | 19.8 |
| TTS data augm. ([Laptev et al., 2020](#)) | - | 6.8 | 19.9 |
| TTS data augm. ([Laptev et al., 2020](#)) | LSTM | **4.3** | **13.5** |
| E2E baseline (ours) | - | 7.4 | 20.1 |
| LaSyn-FixedProj-LFR (ours) | - | 4.5 | 17.1 |
| LaSyn-FixedProj (ours) | - | 4.5 | 16.1 |
| LaSyn-Diffusion (ours) | - | **4.4** | **15.6** |

Table 3: Low-resource ASR results trained with train-clean-100 split of LirbiSpeech. We compare LaSyn with published supervised methods. We report WER (%) on dev/test sets.

pared to the E2E baseline. We notice that the improvement on test-clean is more significant than test-other. Both the fixed-projection latent synthesizer and the diffusion latent synthesizer are trained with train-clean-100 which contains only clean speech. We speculate that the limited variety of training data train-clean-100 biases ASR performance toward clean speech.

We also observe that the performance of the model with fixed-projection latent synthesizer (LaSyn-FixedProj) is only slightly worse than LaSyn-Diffusion. The result is surprising, as the fixed-projection latent synthesizer is simpler than the diffusion latent synthesizer. The diffusion latent synthesizer may need further hyper-parameter tuning, or may need more training data for better performance. The LaSyn-FixedProj-LFR model utilizes a low frame rate speech latent encoder as illustrated in Fig. 5. The performance is slightly worse than the LaSyn-FixedProj on test-other.

Compared to published supervised ASR mod-

| Model | # Params | IC (ACC %) | SF (SLU-F1) |
|---|---|---|---|
| ESPnet-SLU (Arora et al., 2022) | ≥ 300 M | 86.3 | 71.9 |
| PF-hbt-base (Wang et al., 2021) | ≥ 90 M | 87.5 | 75.3 |
| EF-hbt-large (Wang et al., 2021) | ≥ 300 M | **89.4** | 78.4 |
| E2E Baseline (ours) | 37.8 M | 84.4 | 74.7 |
| LaSyn-Diffusion (ours) | 37.8 M | 87.4 | 77.3 |
| LaSyn-FixedProj (ours) | 37.8 M | 88.5 | **78.5** |

Table 4: Results on SLURP dataset. We report accuracy (ACC%) for the IC task and SLU-F1 for the SF task.

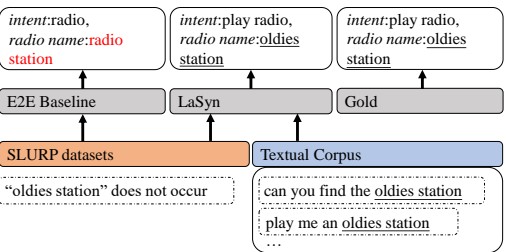

Figure 8: An example of LaSyn output from SLURP test set. The target "oldies station" does not appear in SLU training data while LaSyn utilizes knowledge from the textual corpus. Meta values such as "intent" and slot type are *italicized*.

els that utilize text data through external language models, LaSyn models perform better without an external language model (LM). Compared to the published methods using TTS for data augmentation, the performance of LaSyn models are significantly better without an external LM. Given the existence of real-world scenarios with limited labeled speech data, such as minority languages and specific domains, our proposed method offers a novel approach to developing ASR applications.

## 4.4 SLU Results

### 4.4.1 SLURP

The experimental results of SLURP are shown in Table 4. We report accuracy for intent classification (IC), and SLU-F1(Bastianelli et al., 2020) for slot filling (SF).

We first compare LaSyn models with our E2E baseline. Compared to the E2E baseline, LaSyn-FixedProj improves IC accuracy and SF SLU-F1 by absolute 4.1% and 3.8% respectively. The result suggests that knowledge of textual NLP data is effectively transferred to SLU model. LaSyn-Diffusion performs slightly worse than LaSyn-FixedProj. We believe that with further hyperparameter tuning and more training data, the performance of diffusion latent synthesizer should be further improved.

We further compare the LaSyn models with previously published E2E SLU results. The published models are fine-tuned from HuBERT (Hsu et al., 2021) Base (95 M parameters) or Large (300 M parameters). The performance of LaSyn-FixedProj is comparable to ESPnet-SLU (Arora et al., 2022) and PF-hbt-base (Wang et al., 2021). The IC accuracy of LaSyn-FixedProj is slightly worse than EF-hbt-large (Wang et al., 2021), but the number of parameters is 8 times fewer.

To understand how LaSyn improves our baseline E2E SLU model, we further analyze samples from

the test set that LaSyn performs better than our baseline. An example is shown in Fig. 8. Our E2E baseline model fails for the slot "Oldies Station", as this phrase never occurs in the SLURP training set. In contrast, LaSyn model correctly predicts the slot value. This phrase is included in the textual corpora. The text knowledge is transferred to SLU model with the LaSyn framework. The baseline E2E SLU model does not get the proprietary term 'Oldies Station' while LaSyn predicts this unique vocabulary successfully.

### 4.4.2 STOP

We present our results of STOP in Table 5. Compared to our E2E baseline, LaSyn-FixedProj improves EM accuracy and EM-Tree accuracy on the test set by absolute 4.49% and 2.25% respectively, again suggesting that there is effective cross-modality text knowledge transfer.

We further compare our results with STOP-E2E and STOP-Cascaded (Tomasello et al., 2022). STOP-E2E is an encoder-decoder based Transformer model fine-tuned from an E2E ASR model. The E2E ASR model is fine-tuned from HuBERT Base (Hsu et al., 2021). STOP-Cascaded is a cascaded system composed of an ASR system fine-tuned from wav2vec2.0 Base (Baevski et al., 2020) and an NLU model fine-tuned from a BART Base model (Lewis et al., 2019). LaSyn-FixedProj performs slightly better than STOP-E2E with 0.25% and 2.63% absolute improvement of EM and EM-Tree accuracies on the test set respectively. However, compared to STOP-Cascaded on the test set, while LaSyn-FixedProj is competitive on EM-Tree accuracy, EM accuracy is slightly inferior. The number of parameters in LaSyn models is much fewer. We expect performance improvement with more model parameters.

| Model | # Params | dev EM / EM-Tree | test EM / EM-Tree |
|---|---|---|---|
| STOP-E2E (Tomasello et al., 2022) | ≥ 90 M | 69.12 / 83.89 | 69.23 / 82.87 |
| STOP-Cascaded (Tomasello et al., 2022) | ≥ 230 M | 72.43 / 86.58 | **72.36 / 85.77** |
| E2E Baseline (ours) | 37.8 M | 64.02 / 82.84 | 64.99 / 82.25 |
| LaSyn-Diffusion (ours) | 37.8 M | 67.91 / 85.57 | 68.33 / 84.92 |
| LaSyn-FixedProj (ours) | 37.8 M | 69.33 / 86.24 | **69.48 / 85.50** |

Table 5: Results on STOP dataset. We report the EM and EM-Tree accuracies (%) on dev and test sets.

| Model | Text Data | SLURP (IC / SF) | STOP (dev) (EM / EM-Tree) | STOP (test) (EM / EM-Tree) |
|---|---|---|---|---|
| E2E Baseline | - | 84.4 / 74.7 | 64.02 / 82.84 | 64.99 / 82.25 |
| LaSyn-FixedProj | labelled | 88.5 / 78.5 | 69.33 / 86.24 | 69.48 / 85.50 |
| LaSyn-FixedProj | unlabelled | 86.1 / 75.4 | 66.13 / 82.89 | 66.40 / 82.33 |

Table 6: Ablation study of unlabeled text data. We report results on SLURP test set, STOP dev and test sets.

## 5 Ablation Study

### 5.1 Training with Unlabeled Textual Data

Plain text data without annotation are more abundant than annotated NLP data. We experiment with SLU training with unlabelled textual data. We prepare the unlabelled text data by striping the annotation labels of the NLP datasets and keeping the input text. We apply the MLM task described in section 4.1.2 to utilize the unlabeled textual data. We evaluate LaSyn models with fixed-projection latent synthesizer. The results are listed in Table 6.

The results show that LaSyn still benefits from unlabeled text, compared to our E2E baseline on both SLURP and STOP datasets. With unlabeled text and MLM tasks, LaSyn achieves an absolute improvement of 1.6 % and 0.9 % on IC and SF tasks on SLURP dataset, 2.19 %, and 0.46% on EM and EM-Tree on STOP test set. While the improvement is not as significant as using labeled textual data, data collection is further simplified with unlabelled textual data.

### 5.2 Training with Diverse NLP Tasks

We do an ablation to observe the effect of training LaSyn with textual data from a diverse set of NLP tasks. The results are shown in Table 7. We observe that including each NLP task brings substantial improvement over the E2E baseline. As the NLU task is the most relevant to SLU, performance improvement is the most significant. When we combine all the NLP tasks, there is marginal further performance improvement.

| Model | Text Training data | STOP (dev) (EM / EM-Tree) | STOP (test) (EM / EM-Tree) |
|---|---|---|---|
| E2E baseline | - | 64.02 / 82.84 | 64.99 / 82.25 |
| LaSyn | NLU | 68.99 / 86.31 | 69.40 / 85.45 |
| | NER | 68.55 / 85.65 | 69.24 / 85.05 |
| | IE | 68.43 / 85.50 | 68.88 / 84.99 |
| | NLU + NER + IE | 69.33 / 86.24 | 69.48 / 85.50 |

Table 7: Results of LaSyn trained with text data of different NLP tasks. We report EM and EM-Tree accuracies (%) on STOP dev and test sets.

| Model | SLURP (IC / SF) | STOP (dev) (EM / EM-Tree) | STOP (test) (EM / EM-Tree) |
|---|---|---|---|
| E2E Baseline | 84.4 / 74.7 | 64.02 / 82.84 | 64.99 / 82.25 |
| LaSyn (Acoustic Aug.) | 86.9 / 76.0 | 67.69 / 85.18 | 68.25 / 84.50 |

Table 8: Results of acoustic augmentation with latent synthesizer. We report IC (ACC%) and SF (SLU-F1) for SLURP, EM and EM-Tree accuracies (%) for STOP.

### 5.3 Latent Synthesizer as Acoustic Augmentation

We experiment with using the fixed-projection latent synthesizer for acoustic augmentation. We extract the transcription and the annotation from the SLU dataset to form an NLU dataset. When training the backbone model, we apply both the SLU and the NLU datasets in dual-modality training. As the NLU dataset is derived from the SLU dataset, the latent synthesizer does not introduce extra textual content. Pseudo speech latent representations from the latent synthesizer are considered as an augmentation of the original speech latent representation.

As shown in Table 8, SLU performance improves significantly over the E2E baseline but does not reach the level of Table 7 which utilizes extra NLP datasets. Further enriching the diversity of pseudo acoustic latent is the potential to improve SLU performance.

## 6 Conclusion

We present LaSyn, a framework which enables efficient textual data utilization for E2E speech processing. By converting text into pseudo acoustic latent representation with a latent synthesizer, cross-modality knowledge transfer from textual data to E2E speech processing models is achieved. For the low-resource ASR task with Librispeech, LaSyn achieves relative WER reduction from 22.3% to 40.5% on test sets, compared to our E2E baseline with the same model structure. The results are competitive to published works which utilize textual data through external language models. For SLU

tasks, LaSyn improves over our E2E baseline by absolute 4.1% and 3.8% for for IC accuracy and SF SLU-F1 on SLURP, and absolute 4.49% and 2.25% of EM and EM-Tree accuracies on STOP. The results are competitive to published SOTA works with much fewer model parameters. Future improvement of latent synthesizer should further bridge the gap between speech and textual modality, which we leave as next step.

## Limitations

The core of our method is the generation of pseudo acoustic representation from text input. We focus on generating consistent latent sequences effectively. We only evaluate two latent synthesis methods, including fixed-projection and diffusion latent synthesizers. There are other probable methods for latent generation, such as generative adversarial networks (GAN) (Goodfellow et al., 2020). Compared with TTS which generates audible speech suitable for human judgment, there is no subjective method to evaluate the quality and intelligibility of generated pseudo acoustic representation from the proposed framework, which is a main limitation. The design of reasonable quality indicators of acoustic representation would be meaningful for future work. Moreover, we have not evaluated the proposed latent synthesis framework on other phonological systems such as tonal languages like Chinese. The effectiveness of the framework on tonal languages is not guaranteed.

## Ethics Statement

In this paper, we only use publicly available datasets for experiments. Our experiments do not involve any subjective tests or human data annotations. In the experiments, the latent synthesis framework does not produce any audible speech content. We do not apply any specific speaker information during training and inference.

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

Association for Computational Linguistics.