# OpenReview forum: "Improving End-to-End Speech Processing by Efficient Text Data Utilization with Latent Synthesis"
_EMNLP/2023/Conference — EMNLP 2023 Findings_

### Official Review · Reviewer_tecw · 2023-08-04

**Soundness:** 3

**Excitement:**

4: Strong: This paper deepens the understanding of some phenomenon or lowers the barriers to an existing research direction.

**Missing References:**

I believe the paper would benefit from covering the problem also from the point of view of acoustic->linguistic cross-modal training, such as https://arxiv.org/pdf/2302.03540.pdf. Even if the focus is not on how to properly develop and train speech latent encoders, there is potential interaction and extended impact in covering both literatures.

I also miss references to the recent trends such as SpeechGPT models that are potential competitors (or beneficiaries?) of this approach https://arxiv.org/pdf/2305.11000.pdf

**Paper Topic And Main Contributions:**

This paper proposes a framework for leveraging pure textual data as a way to improve the modeling capability of speech tasks that originally required paired audio-text pairs. This makes it possible to grow the representation capability and accuracy of models by leveraging linguistic information proyected into the acoustic latents. This, the authors hypothesize that promotes cross-modal (linguistic->acoustic) knowledge transfer. Results are demonstrated on both speech recognition and spoken natural language understanding tasks, improving over baselines and meeting state of the art performance on smaller models and (paired) datasets.

**Reasons To Accept:**

This paper proposes an interesting formulation to include unpaired data to speech models. While there has been an abundant recent amount of work to incorporate unscripted audio (audio without text pairs) there has been limited recent work on the 'other side of the coin'. This has been mostly carried out by leveraging pre-trained Language Models, so in this paper the authors propose a lighter approach to use linguistic information in a cross-modal way. This is demonstrated to have a positive impact across multiple downstream tasks, which reinforces the value of linguistic information in multiple speech-related tasks beyond the conventional domains.

**Reasons To Reject:**

Not a strong reason to reject, but I am left wondering if the impact of the proposed approach happens because the latent acoustic representation trained with the datasets is incomplete (in terms, for example, of linguistic coverage). As there is an abundance of speech-text paired datasets in the public domain, it could be that the benefits of the approach are restricted to lower-resource languages in which the technique was untested, leading to a lower impact of the proposed knowledge.

**Reproducibility:**

4: Could mostly reproduce the results, but there may be some variation because of sample variance or minor variations in their interpretation of the protocol or method.

**Reviewer Confidence:**

4: Quite sure. I tried to check the important points carefully. It's unlikely, though conceivable, that I missed something that should affect my ratings.

---

> ### Author Rebuttal · Authors · 2023-08-29
>
> Thanks for your kind and detailed review.
>
> > Not a strong reason to reject, but I am left wondering if the impact of the proposed approach happens because the latent acoustic representation trained with the datasets is incomplete (in terms, for example, of linguistic coverage). As there is an abundance of speech-text paired datasets in the public domain, it could be that the benefits of the approach are restricted to lower-resource languages in which the technique was untested, leading to a lower impact of the proposed knowledge.
>
> In this work, we focus on the low-resource setting as there are real-world scenarios where labeled speech data are still limited, e.g., minority languages and specific domains. For example, the STOP dataset used in our SLU experiment is the largest public SLU dataset so far which contains only 218 hours of paired data.
>
> It is interesting whether the proposed LaSyn method is effective in the high-resource setting and we are considering exploring it in future work.
>
> > Missing References
>
> We will make revisions to the paper to include the missing references you mentioned. We believe that incorporating them into our paper will better support and enrich our perspectives.

---

### Official Review · Reviewer_Paaz · 2023-08-07

**Soundness:** 3

**Excitement:**

2: Mediocre: This paper makes marginal contributions (vs non-contemporaneous work), so I would rather not see it in the conference.

**Missing References:**

・Popuri et al., Enhanced Direct Speech-to-Speech Translation Using Self-supervised Pre-training and Data Augmentation, INTERSPEECH, 2022.

・Siuzdak et al., WavThruVec: Latent speech representation as intermediate features for neural speech synthesis, INTERSPEECH, 2022.

**Paper Topic And Main Contributions:**

This paper presents a novel data augmentation framework aimed at enhancing low-resource automatic speech recognition (ASR) and spoken language understanding (SLU) by leveraging unpaired text data. The authors introduce latent synthesizers, which generate continuous latent features from text. These latents correspond to the output features of the speech encoder, enabling the latent synthesizer to produce simplified representations compared to traditional mel-spectrogram and waveform approaches. The study investigates fixed-projection and diffusion-based models as potential latent synthesizers. Experimental evaluations demonstrate that the proposed augmentation technique leads to substantial improvements in both ASR and SLU quality.

**Reasons To Accept:**

The proposed method is novel to some extent. If the authors conducted more experiments in the speech translation task, the claims would be stronger.

**Reasons To Reject:**

・A comparison with data augmentation methods utilizing discrete acoustic units like SpeechUT is absent. These discrete units simplify speech by emphasizing phonetic and semantic information. An alternative to latent synthesizers involves training Transformer models to generate these discrete units from text [Popuri+ 2022]. While this approach necessitates an additional module to map discrete units to the continuous space, prior research has demonstrated quality enhancements through this technique.

・Comparisons with TTS augmentation are unfair due to the encoder architectures (Transformer vs Conformer). Consequently, the results don't definitively indicate the superior data augmentation method.

・The investigation into the efficacy of external language model fusion is lacking. Since the text data employed in data augmentation originates from the same source as LM training data (specifically Librispeech 960h in this study), we seek clarification on two aspects: 1) the relative effectiveness of the proposed method compared to shallow fusion, and 2) whether the proposed method complements shallow fusion.

・The results in Section 4.2.1 highlight the dependence of data augmentation effectiveness on the domains of training data used for latent synthesizers. This restriction confines the applicability of the proposed method. To investigate domain transferability, employing an ASR model trained on an out-of-domain dataset for initializing the Guiding Net would be desirable.

**Reproducibility:**

4: Could mostly reproduce the results, but there may be some variation because of sample variance or minor variations in their interpretation of the protocol or method.

**Reviewer Confidence:**

5: Positive that my evaluation is correct. I read the paper very carefully and I am very familiar with related work.

---

> ### Author Rebuttal · Authors · 2023-08-29
>
> Thank you for your questions and valuable feedback.
>
> > A comparison with data augmentation methods utilizing discrete acoustic units like SpeechUT is absent. These discrete units simplify speech by emphasizing phonetic and semantic information. An alternative to latent synthesizers involves training Transformer models to generate these discrete units from text [Popuri+ 2022]. While this approach necessitates an additional module to map discrete units to the continuous space, prior research has demonstrated quality enhancements through this technique.
>
> SpeechUT leverages speech pre-training such as HuBERT for speech-to-unit task. The reported results from LaSyn do not utilize speech pre-training but can be initialized with fine-tuned ASR from pre-trained models. The amount of utilized text information is not the same. Therefore the results are not directly comparable. We have already included SpeechUT in the related work and are considering expanding the discussion in the revised edition.
>
> Methods utilizing discrete units, which have been demonstrated quality enhancements, and LaSyn utilizing continuous latent features are both promising and worth exploring.
>
> Methods that utilize discrete units involve discretizing speech signals, which requires the construction of speech-to-unit and text-to-unit generators, e.g., the k-means model learned from HuBERT for speech, and Transformer models for text-to-unit conversion.
>
> In contrast, our LaSyn method directly generates continuous latent features from text by learning a latent synthesizer, e.g., a diffusion model or a fixed projection model.
>
> > Comparisons with TTS augmentation are unfair due to the encoder architectures (Transformer vs Conformer). Consequently, the results don't definitively indicate the superior data augmentation method.
>
> We emphasize that our models are not based on Conformer.
>
> Our models are still based on Transformer even though we add interleaved strided-convolutions between transformer layers for downsampling. We compared our baseline model with Conformer-based ASR and found that the performance is comparable as listed in Table 3. The LaSyn models perform significantly better than our baseline model and Conformer-based ASR.
>
> > The investigation into the efficacy of external language model fusion is lacking. Since the text data employed in data augmentation originates from the same source as LM training data (specifically Librispeech 960h in this study), we seek clarification on two aspects: 1) the relative effectiveness of the proposed method compared to shallow fusion, and 2) whether the proposed method complements shallow fusion
>
> We compared with the published ASR results with LM in Table 3. Our results of LaSyn models without LM are better or comparable. While LM is a validated approach to utilizing text data, decoding with LM introduces additional complexity and computational costs. As our LaSyn model, when performing ASR decoding, is the same as conventional Transducer-based ASR, the model is compatible with shallow fusion and can utilize LM if available.
>
> > The results in Section 4.2.1 highlight the dependence of data augmentation effectiveness on the domains of training data used for latent synthesizers. This restriction confines the applicability of the proposed method. To investigate domain transferability, employing an ASR model trained on an out-of-domain dataset for initializing the Guiding Net would be desirable.
>
> In the ASR experiments, the guiding net is trained only with train-clean-100 split of LibriSpeech. We still observe performance improvement on test-other dataset. Covariate shift is observed in the test-other dataset as they are recorded in a more complex background environment compared with test-clean dataset.
>
> > Missing References
>
> We will make revisions to the paper to include the missing references you mentioned.

---

### Official Review · Reviewer_1c8Z · 2023-08-11

**Soundness:** 4

**Excitement:**

3: Ambivalent: It has merits (e.g., it reports state-of-the-art results, the idea is nice), but there are key weaknesses (e.g., it describes incremental work), and it can significantly benefit from another round of revision. However, I won't object to accepting it if my co-reviewers champion it.

**Missing References:**

Relevant papers on improving/adapting latent speech representation with text only data.

1. Text-Only Domain Adaptation Based on Intermediate CTC. https://www.isca-speech.org/archive/pdfs/interspeech_2022/sato22_interspeech.pdf
2. Integrating Text Inputs For Training and Adapting RNN Transducer ASR Models. https://arxiv.org/pdf/2202.13155.pdf

**Paper Topic And Main Contributions:**

The researchers introduce a framework called LaSyn, designed to efficiently utilize textual data for end-to-end (E2E) speech processing models. This framework facilitates the transfer of cross-modal knowledge from text to E2E speech processing models through latent synthesis. Two implementations of the latent synthesizer are designed as the core of LaSyn: a fixed-projection latent synthesizer and a diffusion latent synthesizer, which incorporates recent advancements in diffusion models. By leveraging LaSyn to enhance E2E speech models with textual data, the researchers achieve competitive results multiple ASR/SLU tasks.

**Reasons To Accept:**

1. Two novel designs of latent synthesizer are tailored to the end goal of text-to-speech representation simulation.
2. The effectiveness of the approach is empirically proven by promising results on multiple tasks/datasets.
3. Good paper writing and clear presentation.

**Reasons To Reject:**

1. The ASR experiments are limited. Librispeech is a widely used dataset, but there are various more competitive models (e.g. Wave2Vec2, HuBERT etc) than the baselines in the paper. Moreover, since Librispeech is clean read speech data and not a very challenging dataset, experimenting on it alone may not reflect the ASR performance comprehensively.
2. When targeting low-resource languages/domains, the availability of G2P are also be limited (section 3.2.2).

**Reproducibility:**

4: Could mostly reproduce the results, but there may be some variation because of sample variance or minor variations in their interpretation of the protocol or method.

**Reviewer Confidence:**

4: Quite sure. I tried to check the important points carefully. It's unlikely, though conceivable, that I missed something that should affect my ratings.

---

> ### Author Rebuttal · Authors · 2023-08-29
>
> Thank you for your valuable feedback and comments. We next provide explanations for your concerns.
>
> > The ASR experiments are limited. Librispeech is a widely used dataset, but there are various more competitive models (e.g. Wave2Vec2, HuBERT etc) than the baselines in the paper.
>
> Our method is complementary to speech pre-training as LaSyn is initialized with a Transformer-based ASR which can be fine-tuned from a pre-trained model.
> The official result (WER) of Wav2Vec 2.0 BASE in low-resource ASR experiment is 6.1\% / 13.3\% for test-clean / other without external LM. While Wav2Vec 2.0 utilizes 960-hour unlabelled speech for pre-training, our LaSyn framework in contrast utilizes 960-hour text transcription (4.4\% / 15.6\%). Both frameworks exhibit their own strength on test clean / other respectively.
>
> > Moreover, since Librispeech is clean read speech data and not a very challenging dataset, experimenting on it alone may not reflect the ASR performance comprehensively.
>
> We acknowledge that testing on more varieties of ASR corpora for ASR task would be more comprehensive. We consider ASR task as the first stage of traditional pipe-lined SLU system. We include the ASR task with LibriSpeech 100-hour data to evaluate the soundness of LaSyn design on basic speech processing. We further conducted more comprehensive experiments on SLU tasks and demonstrated the effectiveness of our LaSyn framework.
>
> > When targeting low-resource languages/domains, the availability of G2P are also be limited (section 3.2.2).
>
> We acknowledge the importance of addressing the G2P availability in low-resource settings. We will incorporate this perspective in the Limitation section in the revised manuscript. We note that a G2P model can built with various methods other than deep learning, as long as pronunciation dictionaries of the low-resource languages are available.
>
> > Reproducibility: 2: Would be hard pressed to reproduce the results. The contribution depends on data that are simply not available outside the author's institution or consortium; not enough details are provided.
>
> All datasets for experiments in this paper are publicly available. We will release the source code along with the parameters in the Appendix once accepted.
>
> > Missing References
>
> We will make revisions to the paper to include the missing references you mentioned. We believe that incorporating them into our paper will better support and enrich our perspectives.

---

### Meta-Review · Area_Chair_GTY3 · 2023-09-19

**Recommendation:** 4

**Metareview:**

This paper presents a lightweight framework to utilize text for training speech processing models. In particular, it adds a latent synthesizer module that maps text into speech embedding for training a shared backbone model that takes this embedding as input. Two implementations (fixed projection and diffusion) of the latent synthesizers are presented. Effectiveness of the approach is verified on speech recognition (Librispeech) and spoken language understanding (SLURP and STOP).

All reviewers find the approach novel and can be easily reproduced. The proposed approach also demonstrates strong performance over supervised baselines and other semi-supervised baselines on multiple speech processing tasks.

The main caveat of the manuscript is the lack of comparison with semi-supervised baselines that utilizes text data (only compared with TTS data augm. (Laptev et al., 2020)). While I agree with the authors in the rebuttal that SpeechUT may not be a suitable baseline because it is built upon HuBERT utilizing unpaired speech, there are still prior studies such as [1,2]. Specifically, [1] trains a text-to-encoder (TTE) model that predicts ASR embeddings from text and uses that for training shared components. This is highly related to the proposed method, but I acknowledge that proposed methods does have novelty by leveraging diffusion model.

[1] Hayashi, Tomoki, et al. "Back-translation-style data augmentation for end-to-end ASR." 2018 IEEE Spoken Language Technology Workshop (SLT). IEEE, 2018.
[2] Liu, Alexander H., Hung-yi Lee, and Lin-shan Lee. "Adversarial training of end-to-end speech recognition using a criticizing language model." ICASSP 2019-2019 IEEE International Conference on Acoustics, Speech and Signal Processing (ICASSP). IEEE, 2019.

---

### Decision · Program_Chairs · 2023-10-07

**Decision:**

Accept-Findings

**Comment:**

This paper presents a lightweight framework to utilize text for training speech processing models. In particular, it adds a latent synthesizer module that maps text into speech embedding for training a shared backbone model that takes this embedding as input. Two implementations (fixed projection and diffusion) of the latent synthesizers are presented. Effectiveness of the approach is verified on speech recognition (Librispeech) and spoken language understanding (SLURP and STOP).

All reviewers find the approach novel and can be easily reproduced. The proposed approach also demonstrates strong performance over supervised baselines and other semi-supervised baselines on multiple speech processing tasks.

The main caveat of the manuscript is the lack of comparison with semi-supervised baselines that utilizes text data (only compared with TTS data augm. (Laptev et al., 2020)). While I agree with the authors in the rebuttal that SpeechUT may not be a suitable baseline because it is built upon HuBERT utilizing unpaired speech, there are still prior studies such as [1,2]. Specifically, [1] trains a text-to-encoder (TTE) model that predicts ASR embeddings from text and uses that for training shared components. This is highly related to the proposed method, but I acknowledge that proposed methods does have novelty by leveraging diffusion model.

[1] Hayashi, Tomoki, et al. "Back-translation-style data augmentation for end-to-end ASR." 2018 IEEE Spoken Language Technology Workshop (SLT). IEEE, 2018.
[2] Liu, Alexander H., Hung-yi Lee, and Lin-shan Lee. "Adversarial training of end-to-end speech recognition using a criticizing language model." ICASSP 2019-2019 IEEE International Conference on Acoustics, Speech and Signal Processing (ICASSP). IEEE, 2019.